# *Π*-theorem generalization of the ice-age theory

*Mikhail Y. Verbitsky[1] and Michel Crucifix[2]*

[1] Gen5 Group, LLC, Newton, MA, USA
[2]UCLouvain, Earth and Life Institute, Louvain-la-Neuve, Belgium
Correspondence: Mikhail Verbitsky (verbitskys@gmail.com)

**Abstract.**

Analyzing a dynamical system describing the global climate variations requires, in principle, exploring a large space spanned by the numerous parameters involved in this model. Dimensional analysis is traditionally employed to deal with equations governing physical phenomena to reduce the number of parameters to be explored, but it does not work well with dynamical ice-age models, because, as a rule, the number of parameters in such systems is much larger than the number of independent dimensions. Physical reasoning may however allow us to reduce the number of effective parameters and apply dimensional analysis in a way that is insightful. We show this with a specific ice-age model (Verbitsky et al, 2018) which is a low-order dynamical system based on ice-flow physics coupled with a linear climate feedback. In this model, the ratio of positive-to-negative feedback is effectively captured by a dimensionless number called the "*V*-number", which aggregates several parameters and, hence, reduces the number of governing parameters. This allows us to apply the central theorem of the dimensional analysis, the π-theorem, efficiently. Specifically, we show that the relationship between the amplitude and duration of glacial cycles is governed by a property of scale invariance that does not depend on the physical nature of the underlying positive and negative feedbacks incorporated by the system. This specific example suggests a broader idea, that is, the scale invariance can be deduced as a general property of ice age dynamics, if the latter are effectively governed by a single ratio between positive and negative feedbacks.

## 1.  Introduction.

Mathematical modeling of Pleistocene ice ages using astronomically forced spatially-resolving models of continental ice sheets, the ocean, and the atmosphere has always been, and remains a computational challenge. Therefore, though higher resolution models (e.g., Abe-Ouchi et al, 2013) and models of intermediate complexity (e.g., Verbitsky and Chalikov, 1986, Chalikov and Verbitsky, 1990, Gallée et al., 1991, Ganopolski et al, 2010) are gaining popularity, it has been argued for a long time that significantly less computationally demanding dynamical models may provide just as much insight as the models with more degrees of freedom (Saltzman, 1990). However, even though the computational load for solving dynamical equations is minimal, the work and number of experiments needed for spanning the full parameter space is easily overwhelming. Analyzing a dynamical system of ice ages is thus, in principle, a difficult task. In mathematical physics, the method of dimensional analysis (e.g., Barenblatt, 2003) has been traditionally employed to take advantage of symmetry or invariance principles and, as a result, to reduce the number of effective parameters.  It has not been applied to low-order models of the Pleistocene climate, because in such models the number of governing parameters is much larger than the number of independent dimensions. Indeed, the number of independent dimensions in a dynamical system does not exceed the number of variables (it may be smaller if some variables have the same or dependent dimensions), to which one adds time, which is always present in a dynamical system. For example, the dynamical system of Saltzman and Verbitsky (1993) described the evolution of 4 variables: ice volume $(m^3)$, $CO_2$ concentration (ppm), ocean temperature ($^o$C), and bedrock depression (m). The number of independent dimensions, including time, was thus 4. This system had 18 parameters, including the amplitude and the period of the external forcing. In such case, the π-theorem (Buckingham, 1914) — the tenet of dimensional analysis — is of little help to simplify the analysis and effectively provide physical insight, because, even in the dimensionless form, the system would still contain 14 (18 – 4) dimensionless groups.

In Verbitsky et al (2018), we derived a dynamical model of the Pleistocene climate from the scaled conservation equations of viscous non-Newtonian ice, and combined them with an equation describing the evolution of the climate temperature. The work was motivated by the prospect of delivering a low-order, parsimonious approach to the problem of understanding glacial-interglacial cycles. The state of the ice-

climate system is summarized by a 3-dimensional vector: glaciation area $S$ (m$^2$), ice sheet basal temperature $\theta$ ($^{\circ}$C), and climate temperature $\omega$ ($^{\circ}$C). The number of independent dimensions, including time, is thus 3. However, despite our effort to be parsimonious in the physical description, the model includes 12 parameters, which is still much larger than the number of independent dimensions. As now we may have 9 (12 – 3) dimensionless groups, this is an obvious progress relative to the Saltzman and Verbitsky (1993) model, but not enough for an effective use of the $\pi$-theorem. The situation changed dramatically when we discovered that the dynamical properties of the system are largely defined by the dimensionless $V$-number incorporating 8 model parameters and measuring the ratio of climate positive feedback over the ice sheet's own negative feedback. At once, 7 parameters are effectively eliminated, and using the $\pi$-theorem became an attractive prospect. We first applied the $\pi$-theorem reasoning to investigate the propagation of millennial forcing into ice-age dynamics (Verbitsky et al, 2019a) and found that the millennial forcing introduces a disruption, i.e., shifts the system equilibrium point, and this disruption is proportional to the second degree of the forcing period.

In this paper we will apply this approach systematically to all model variables. This will allow us to demonstrate that, in the model, glacial area and climate temperature are scale invariant in the orbital frequencies domain (in the case of the climate temperature – even beyond this domain), and observe that this property does not depend on the specific physical nature of the climate system feedbacks. This observation is important. The empirical analysis of paleoclimate series shows that there is a rich spectral content and points to the existence of "spectral slopes" (e.g., Huybers and Curry, 2006, Lovejoy and Schertzer, 2013). Lovejoy and Schertzer (2013) evoke some generic process, such as the principle of "cascades" which is tightly linked to the concept of scale invariance of the equations. For example, the scale invariance of fluid-dynamics equations is exploited to provide inferences about spectral slopes of turbulent flows. However, to our knowledge, there is no available theory supporting scale invariance in regimes associated with glacial-interglacial dynamics. Yet, paleoclimate simulations with more sophisticated models, including the seminal paper by Abe-Ouchi et al (2013) and the simulations with CLIMBER provided by Ganopolski et al (2010), tend to focus on the response of the ice-sheet climate system to orbital forcing, and discuss the respective amplitudes of the 100-kyr, 41-kyr, and 21-23-kyr periods, but none discuss the slope of the power spectrum down to the millennium scale. Therefore, we believe that our research will provide at least some important elements that should help us to bridge both approaches

Accordingly, our paper is structured as follows. First, we will briefly recapture equations, parameters, and dimensions of the Verbitsky et al (2018) model. Then we will remind the essence of the $\pi$-theorem, apply it to all model variables, and discuss its implications.

## 2.   A dynamical model of Pleistocene glacial rhythmicity.

The non-linear dynamical model of the global climate system (Verbitsky et al, 2018) is derived from the scaled equations of ice sheet thermodynamics, combined with a linear feedback equation involving an effective "temperature", which describes the climate state outside the ice region.

$$\frac{dS}{dt} = \frac{4}{5}\zeta^{-1}S^{3/4}(a - \varepsilon F_S - \kappa\omega - c\theta) \tag{1}$$

$$\frac{d\theta}{dt} = \zeta^{-1}S^{-1/4}(a - \varepsilon F_S - \kappa\omega)\{\alpha\omega + \beta[S - S_0] - \theta\} \tag{2}$$

$$\frac{d\omega}{dt} = \gamma_1 - \gamma_2[S - S_0] - \gamma_3\omega \tag{3}$$

The model variables and their dimensions are defined as follows: $S$ (m$^2$) is the glaciation area, $\theta$ ($^{\circ}$C) is the basal ice sheet temperature, and $\omega$ ($^{\circ}$C) is the effective global climate temperature. The third equation implicitly accounts for the effect of the response of $CO_2$-concentration, along with other radiative feedbacks.

Model parameters along with their dimensions are: $\zeta$ (m$^{1/2}$) is the "shape" factor of the ice sheet; $a$ (m/s) is the characteristic rate of snow precipitation; $F_S$ is normalized mid-July insolation at 65°N (Berger and Loutre, 1991); $\varepsilon$ (m/s) is the amplitude of the external forcing; $\kappa$ (m s$^{-1}$ $^{\circ}$C$^{-1}$) and $c$ (m s$^{-1}$ $^{\circ}$C$^{-1}$) are sensitivity parameters, describing, correspondingly, climate temperature and basal sliding impacts into ice-sheet mass balance; the dimensionless coefficient $\alpha$ describes basal temperature sensitivity to global climate temperature changes, coefficient $\beta$ ($^{\circ}$C /m$^2$) defines basal temperature dependence on ice sheet

dimensions, $S_0$ (m$^2$) is a reference glaciation area; $\gamma_1$ ($^o$C/s), $\gamma_2$ ($^o$C m$^{-2}$ s$^{-1}$) and $\gamma_3$ (s$^{-1}$) define climate temperature evolution, $1/\gamma_3$, being a time constant. If the forcing is periodic, then we may consider that the system dynamics is described by an additional parameter: the forcing period $T$ (s). Thus we have a system of 3 variables, 3 (including time) independent dimensions, and 12 parameters. The system (1) – (3) is not sensitive to initial conditions and, therefore, we do not include the latter into the list of parameters.

Physical reasoning and numerical experiments (Verbitsky et al., 2018) led us to the suggestion that the system response is essentially determined by the $V$-number measuring a balance between positive and negative model feedbacks:

$$V = \frac{1}{\beta}\left(\alpha + \frac{\kappa}{c}\right)\left(\frac{\gamma_2}{\gamma_3} - \frac{\gamma_1}{S_0\gamma_3}\right) \tag{4}$$

Here parameter $\beta$ is a measure of ice-sheet negative feedback. The term $(\alpha+\kappa/c)(\gamma_2/\gamma_3 - \gamma_1/\gamma_3/S_0)$ measures the climate system positive feedback (Verbitsky et al, 2018).

If we assume that the $V$-number effectively captures the behavior of the model with respect to the 8 parameters included in its definition, then the number of parameters is effectively reduced to 5: $V$, $\zeta$, $a$, $\varepsilon$, and $T$. We assume further that parameter $\zeta$ in equations (1) - (2) is a constant, thus assuming an invariant relationship between ice thickness $H$ and glaciation area $S$ ($H = \zeta S^{1/4}$, Verbitsky et al, 2018). We also note that the $V$-number has been assembled using components of the steady-state solution of the system (1) – (3) (Verbitsky et al, 2018). Obviously, parameter $\zeta$, as a multiplier, is not part of this steady-state solution. Therefore our hypothesis that the $V$-number defines the model's behavior, in fact also includes the assumption that the impact of the parameter $\zeta$ on the system behavior, at the reference value, is weak. As a result, we end up with the assumption that the system's response to external forcing is essentially determined by no more than four parameters: $V$, $a$, $\varepsilon$, and $T$. We will now learn how to take profit of this advantage.

## 3. Dimensional analysis of model variables.

### 3.1 Period of the system response to the external forcing, $P$.

We previously noticed (Verbitsky et al, 2018), that with weak climate positive feedback ($V \sim 0$), the system, exhibits fluctuations in response to the astronomical forcing with a dominating period of about 40 kyr, which may arise either as direct response to obliquity, or as a doubled-period response to the forcing associated with climatic precession (2 x 20 kyr). When the climate positive feedback intensifies such that $V \sim 0.75$ and external forcing is strong, the system evolves with a doubled obliquity period. We can therefore assume that the period of the system response to the external forcing, $P$, is a function of the $V$-number, the amplitude of the external forcing, $\varepsilon$, and of the period of the external forcing, $T$. We thus begin with the most general hypothesis:

$$P = \psi(V, a, \varepsilon, T) \tag{5}$$

This is at this stage that the $\pi$-theorem intervenes. Specifically, it stipulates that a physical relationship should not depend on a system of units and therefore, in the dimensionless form, the number of dimensionless arguments is equal to the total number of the governing parameters minus the number of governing parameters with independent dimensions (Buckingham, 1914). If we select dimensions of $\varepsilon$ and $T$ as independent dimensions, then application of the $\pi$-theorem to the equation (5) gives us:

$$P/T = \Psi(V, \varepsilon/a) \tag{6}$$

$$P = T\Psi(\Pi_1, \Pi_2), \Pi_1, = V, \Pi_2 = \varepsilon/a \tag{7}$$

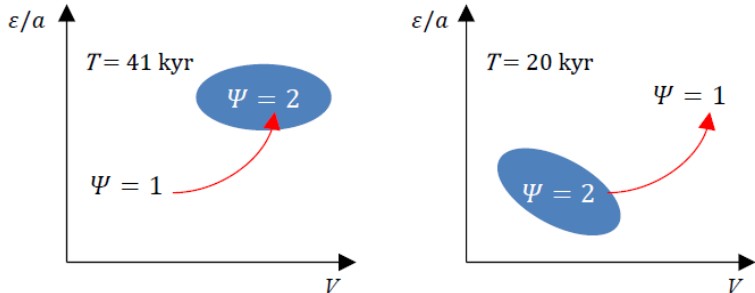

Fig. 1. A typical illustrative $\Psi(V, \varepsilon/a)$ function. Red arrow represents hypothetical trajectory of the
system's Pleistocene history: from doubled precession periods of the early Pleistocene to doubled obliquity
periods of the late Pleistocene.

Fig. 1 presents a sketch of how the function $\Psi(V, \varepsilon/a)$ may look like, qualitatively. The underlying idea is
that the Pleistocene history of the climate system may be understood as a trajectory in the $[V, \varepsilon/a]$ space
(Crucifix and Verbitsky, 2019). The shape and location of the period doubling domain $\Psi = 2$ is expected to
depend on the forcing period.
It is interesting that Figure 1 is consistent with a similar map produced by a conceptual model built on
a completely different principle, i.e., the simple oscillator type model of Daruka and Ditlevsen (2016). In
both cases, the obliquity period doubling requires relatively intense external forcing in combination with
the relatively high $V$-number (or reduced damping in the case of Daruka and Ditlevsen, 2016). This
similarity implies that the importance of the $V$-number for a climate system dynamics may extend well
beyond the Verbitsky et al (2018) model.

**3.2  Amplitude of the glacial area variations, $\acute{S}$.**
We begin again with the most general hypothesis. We suggest that the amplitude of glacial area variations
$\acute{S}$ is a function of the $V$-number, of the characteristic rate of snow precipitation, $a$, of the amplitude of the
external forcing $\varepsilon$, and of the period of the system response $P$ as it is described by equation (7). The
relationship between the period of the response and that of the forcing may therefore be non-trivial. It
means that the system response may exhibit original forcing periods or multiples of them.
$$\acute{S} = \varphi(V, a, \varepsilon, P) \tag{8}$$
If the hypothesis (8) is true, then, taking dimensions of $\varepsilon$ and $P$ as independent dimensions, and using the $\pi$-
theorem, we obtain:
$\acute{S}/(\varepsilon^2 P^2) = \Phi(V, \varepsilon/a)$ , and finally:
$$\acute{S} = \varepsilon^2 P^2 \Phi(\Pi_1, \Pi_2) \tag{9}$$
Neither $\Pi_1$ nor $\Pi_2$ contain $P$. Equation (9) therefore implies that, at constant amplitude of the external
forcing $\varepsilon$, the amplitude of glacial area variations is scale invariant with a frequency slope equal 2. Fig. 2
($\acute{S}$, reference parameters values) presents a numerical test of the hypothesis (8) and of its implication (9).
Here, we measure the system response to single-sinusoid forcings of constant amplitude and periods $T$
varying from 5 kyr to 50 kyr. The system responds to this forcing with periods $P$ ranging from 5 kyr to 100
38  kyr, because forcing periods $T$ of 40 kyr and 50 kyr produce response periods $P$ of 80 kyr and 100 kyr,
correspondingly. It can be seen that the $\acute{S}$-amplitude frequency slope, $\beta_a$, is close to 2 (i.e., $\beta_a = 1.8$) for
periods between 30 ky and 100 ky. It means that the *amplitude of glacial area variations is scale invariant*
*in the orbital domain.*

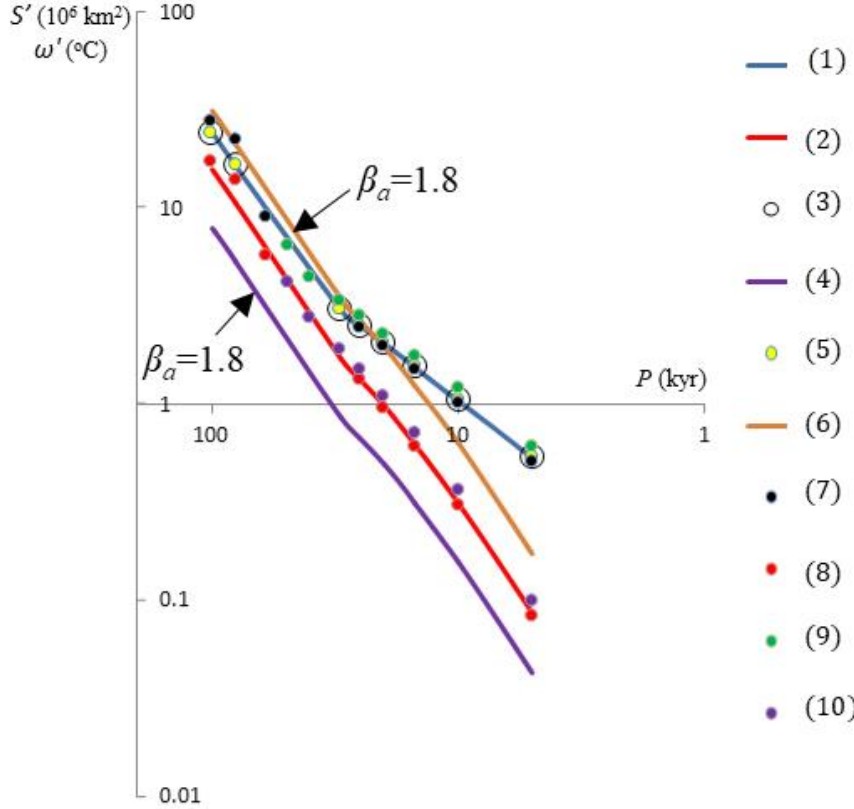

**Fig. 2** The system response to a single-sinusoid external forcing of constant amplitude and different
periods: (1) $\acute{S}$, reference parameters values; (2) $\acute{\omega}$, reference parameters values; (3) $\acute{S}$, intensive climate
temperature and weak albedo positive feedbacks; (4) $\acute{\omega}$, intensive climate temperature and weak albedo
positive feedbacks; (5) $\acute{S}$, weak climate temperature and intensive albedo positive feedbacks; (6) $\acute{\omega}$, weak
climate temperature and intensive albedo positive feedbacks; (7) $\acute{S}$, intensive climate temperature positive
and ice-sheet basal temperature negative feedbacks; (8) $\acute{\omega}$, intensive climate temperature positive and ice-
sheet basal temperature negative feedbacks; (9) $\acute{S}$, weak climate temperature positive and ice-sheet basal
temperature negative feedbacks; (10) $\acute{\omega}$, weak climate temperature positive and ice-sheet basal temperature
negative feedbacks.

**3.3  Amplitude of the basal temperature, $\acute{\theta}$.**
The amplitude spectrum of the $\theta$-variable cannot be derived unambiguously from the same simple

considerations as we have employed for $P$ and $\acute{S}$  because: (a) we cannot constrain ourselves with only

parameters $V$, $a$, $\varepsilon$, and $P$, since the basal temperature $\theta$ is measured in $^{o}$C, but neither $\varepsilon$, nor $a$, nor $P$

contain $^{o}$C, but (b) as soon as we disassemble the $V$-number, i.e., use all individual model parameters

instead of $V$, the advantage of using the $\pi$-theorem is lost. Nevertheless, if we disassemble the $V$-number

wisely, we can minimize the number of dimensional parameters and, as a result, we may be rewarded by

discovering the identities of critical groups that define the scaling properties of $\theta$. Accordingly, we will

disassemble the $V$-number using not individual parameters involved but, instead, using dimensionless

groups that are present in the $V$: $\alpha, \dfrac{\kappa}{c}, \dfrac{\gamma_1}{\beta\gamma_3 S_0}, \dfrac{\gamma_2}{\beta\gamma_3}$. If we consider that the group $\dfrac{\gamma_1}{\beta\gamma_3 S_0}$ is a dimensionless

representation of the parameter $\gamma_1$, and the group $\frac{\gamma_2}{\beta\gamma_3}$ is a dimensionless representation of the parameter $\beta$, then the remaining parameters $\gamma_2, \gamma_3, S_0$ need to be represented individually in the dimensional form.

Taking this together, this yields the following hypothesis:

$$\acute{\theta} = \chi\left(\alpha, \frac{\kappa}{c}, \frac{\gamma_1}{\beta\gamma_3 S_0}, \frac{\gamma_2}{\beta\gamma_3}, \gamma_2, \gamma_3, S_0, a, \varepsilon, P\right) \tag{10}$$

Taking $\gamma_2, S_0$ and $P$ as independent dimensions, the $\pi$-theorem implies:

$$\acute{\theta} = \gamma_2 S_0 P X\left(\alpha, \frac{\kappa}{c}, \frac{\gamma_1}{\beta\gamma_3 S_0}, \frac{\gamma_2}{\beta\gamma_3}, \gamma_3 P, aPS_0^{-1/2}, \varepsilon PS_0^{-1/2}\right) \tag{11}$$

or, combining groups $\alpha, \frac{\kappa}{c}, \frac{\gamma_1}{\beta\gamma_3 S_0}, \frac{\gamma_2}{\beta\gamma_3}$ back into $V$-number:

$$\acute{\theta} = \gamma_2 S_0 P X\left(V, \gamma_3 P, aPS_0^{-1/2}, \varepsilon PS_0^{-1/2}\right) \tag{12}$$

Since $\Pi_2 = \varepsilon/a$,

$$\acute{\theta} = \gamma_2 S_0 P X(\Pi_1, \Pi_2, \Pi_3, \Pi_4) \tag{13}$$

where $\Pi_3 = \gamma_3 P, \Pi_4 = \varepsilon PS_0^{-1/2}$.

As $\Pi_3$ and $\Pi_4$ include $P$, then, generally speaking, *the amplitude of basal temperature variations is not expected to be scale invariant.* Under some circumstances though, the function $X(\Pi_1, \Pi_2, \Pi_3, \Pi_4)$ may become $P$-independent and the amplitude of the basal temperature variations may develop the property of scale invariance. For example, we observed experimentally that when $\Pi_4 \to 0$ (e.g., the amplitude of the external forcing, $\varepsilon$, is reduced), the equation (13) becomes scale invariant with a frequency slope equal 1. In this case $\acute{\theta} = \gamma_2 S_0 P X(\Pi_1, \Pi_2, \Pi_3/\Pi_4)$.

### 3.4 Amplitude of the climate temperature, $\acute{\omega}$.

Since equation (3) for $\omega$ is linear, it may provide us with a hint about the response scaling characteristics of this variable. In the orbital domain, $\Pi_3 = \gamma_3 P \gg 1$, so that equation (3) may be approximated to: $\gamma_3 \omega \approx \gamma_1 - \gamma_2(S - S_0)$. Hence, $\acute{\omega} = \frac{\gamma_2}{\gamma_3}\acute{S}$. We may hypothesize therefore that in the orbital domain and possibly even beyond:

$$\acute{\omega} = \nu\left(V, \frac{\gamma_2}{\gamma_3}, a, \varepsilon, P\right) \tag{14}$$

Taking the dimensions of $\frac{\gamma_2}{\gamma_3}, \varepsilon$, and $P$ as independent and applying again $\pi$-theorem reasoning, we should expect that:

$$\acute{\omega} = \frac{\gamma_2}{\gamma_3}\varepsilon^2 P^2 N(\Pi_1, \Pi_2) \tag{15}$$

At constant amplitude of the external forcing $\varepsilon$, equation (15) implies that the amplitude of climate temperature variations $\acute{\omega}$ grows with the square of the response period. The results presented in Fig.2 ($\acute{\omega}$, reference parameters values) support the hypothesis (14) and its implication (15): The $\omega$-variable amplitude frequency slope is close to 2 (i.e., $\beta_a = 1.8$) for periods between 5 kyr and 100 kyr. It means that in the orbital and millennial domains, the *amplitude of the climate temperature is scale invariant.*

## 4. Discussion

### 4.1 Scale invariance and a physical nature of the climate system feedbacks

So far, we based our implications of scaling relationships on the significance of a dimensionless number (in our case, the $V$-number) quantifying a mean ratio between positive and negative feedbacks. That is, the scaling relationships found should be robust across changes in the composition of $V$, provided that the value of $V$ is unchanged. To illustrate this implication, we conducted four numerical experiments. In the first experiment, we increase coefficients $\alpha$ and $\kappa$ two-fold and reduce $\gamma_2$ by a half relative to their reference values. This does not change the reference value of the $V$-number (see the equation (4) and note that the reference value of $\gamma_1 = 0$), that is $V=0.75$, but transforms system (1) – (3) to a system where the positive feedback is dominated by the climate temperature affecting ice-sheet mass balance and its temperature regime. We then measure the system response to the single-sinusoid forcing of the same amplitude and periods $T = 5$ - 50 kyr. (Note, that periods $T = 40$ kyr and 50 kyr produce system response of periods $P = 80$ kyr and 100 kyr, correspondingly). In the second experiment, we decrease coefficients $\alpha$ and $\kappa$ by 50% and increase $\gamma_2$ two-fold relative to their reference values. Again, this does not change the reference value of the $V$-number, $V=0.75$, but transforms system (1) – (3) to a system where the positive feedback is dominated by the albedo feedback. In the third experiment, we increase coefficients $\alpha$ and $\kappa$ by 50% as well as the coefficient $\beta$, thus creating the system with intensive climate-temperature positive feedback and intensive ice-sheet basal temperature negative feedback, the $V$-number still being equal to 0.75. And finally, we decrease coefficients $\alpha$, $\kappa$, and $\beta$ by 50%, making a system with weak climate-temperature positive and ice-sheet basal temperature negative feedbacks. The response of all four systems to the external forcing is shown in Fig.2. Despite different underlying physics, all four systems demonstrate the same: in the orbital domain, their amplitudes of glacial area variations are scale invariant with "1.8" frequency slope, and the amplitudes of the climate temperature are scale invariant in the orbital and millennial domains with the same slope.

This robustness is comforting. As we know, the physical interpretation of a low-order dynamical model can be partly ambiguous. For example, the mechanisms responsible for the changes in the "effective climate temperature", and how it impacts the ice mass balance are not fully described in this model. It is therefore reassuring to have been able to identify what seems to be the key ingredient for the scaling relationship, in this case, that a single quantity (the $V$-number) grossly determines the dynamics of the system response. In other words, it relies on the fact that the number of effective parameters is smaller than is apparent from a more detailed description of the system.

This, incidentally, shows how difficult it is to disambiguate the physical mechanisms responsible for a given behavior. Different assemblages yielding the same $V$-number will, indeed, produce slightly different solutions, but less different than one could have perhaps expected. The dimensionless functions like, for example, function $\Phi(V, \varepsilon/a)$ in the equation (9),

$$\acute{S} = \varepsilon^2 P^2 \Phi(V, \varepsilon/a)$$

and function $\Phi'(V, \varepsilon/a)$ corresponding to the same value of the $V$-number but formed by the different physics (different set of parameters),

$$\acute{S} = \varepsilon^2 P^2 \Phi'(V, \varepsilon/a)$$

though are not identical, yield the same scaling behavior. If the amplitude of the external forcing $\varepsilon$ is constant, the period $P$ shows up only as a power-law monomial $\sim P^n$ and its power $n$ makes the same scale-invariant amplitude-spectrum slope *regardless of the specific physics defining the V-number*. In other words, though the functions $\psi(V, a, \varepsilon, T)$, $\varphi(V, a, \varepsilon, P)$, $\chi(V, a, \varepsilon, P)$, and $\nu\left(V, \frac{\gamma_2}{\gamma_3}, a, \varepsilon, P\right)$ may change depending on the specific physics forming the $V$-number, their governing parameters always remain the same because they are determined by the structure of the system (1) – (3). Accordingly, the functions $\Psi(\Pi_1, \Pi_2)$, $\Phi(\Pi_1, \Pi_2)$, $X(\Pi_1, \Pi_2, \Pi_3, \Pi_4)$, and $N(\Pi_1, \Pi_2)$ may also change, but their dimensionless arguments ($\Pi$-groups) remain unaffected. As long as their groups, like, for example, $\Pi_1$ and $\Pi_2$, do not

contain $P$, we have a possibility of scale-invariance. This observation makes the scale invariance a very general and expected property of the climate system.

The physical interpretation of the dynamical model we employ in this study (Verbitsky et al, 2018) is very straightforward as far as equations (1) and (2) are concerned: these are scaled equations of mass and energy conservation of viscous ice flow. We must admit, though, that equation (3) of the "climate temperature" is, indeed, ambiguous. In other words, we are uncertain about some key mechanisms that we have chosen to describe using the "rest-of-the-climate" linear equation. Among others, these may be non-linear effects related to the carbon cycle, non-linear effects of sea-level destabilization of ice sheets and related synchronization, non-linear effects associated with atmospheric circulation, or non-linear effects related to biogenic calcifiers and their action on alkalinity, etc. A challenger might thus claim that these effects are so important that they should be considered more explicitly. Indeed, we have the hope that even after accounting for these processes, we might end up with a model that still has grossly the same mathematical structure as the Verbitsky et al (2018) model, even though the meaning of some of the variables will have changed. Specifically, since equation (3) is linear, it can be split into several equations:

$$\omega = \omega_1 + \omega_2 + \cdots + \omega_n$$

$$\frac{d\omega_1}{dt} = \gamma_{11} - \gamma_{21}(S - S_0) - \gamma_3 \omega_1$$

$$\frac{d\omega_2}{dt} = \gamma_{12} - \gamma_{22}(S - S_0) - \gamma_3 \omega_2$$

$$\dots$$

$$\frac{d\omega_n}{dt} = \gamma_{1n} - \gamma_{2n}(S - S_0) - \gamma_3 \omega_n$$

Each of the above equations may represent different feedback mechanisms. Therefore our experiments with increased (or reduced) $\gamma_2$ may be also understood as experiments with additional feedbacks of different nature ($\gamma_2 = \gamma_{21} + \gamma_{22} + \dots + \gamma_{2n}$), though of the same time-scale $1/\gamma_3$.

### 4.2 Multi-sinusoid forcing

Thus far we assumed a single-sinusoid external forcing with an amplitude $\varepsilon$ and a period $T$. When we force our system with normalized mid-July insolation at 65°N (Berger and Loutre, 1991), this assumption is not valid any longer because both the amplitudes and the periods of precession and obliquity are different. Therefore, the hypothesis (5) must be re-written as:

$$P = \psi[V, a, \varepsilon_1, T_1, \varepsilon_2, T_2] \tag{16}$$

Here $P$ is a period of the system response to a specific forcing component (a peak of the response spectrum), index "1" corresponds to obliquity, and index "2" corresponds to precession. Taking dimensions of $\varepsilon_1$ and $T_1$ as independent dimensions, and using the $\pi$-theorem, we obtain:

$$P_1 = T_1 \Psi_1[V, \varepsilon_1/a, \varepsilon_1/\varepsilon_2, T_1/T_2] \tag{17}$$

Here $P_1$ is a period of the system response to the obliquity forcing. Similarly, taking dimensions of $\varepsilon_2$ and $T_2$ as independent dimensions, and using the $\pi$-theorem, we have:

$$P_2 = T_2 \Psi_2[V, \varepsilon_2/a, \varepsilon_1/\varepsilon_2, T_1/T_2] \tag{18}$$

Here $P_2$ is a period of the system response to the precession forcing. Since in the case of the orbital forcing $\varepsilon_1/\varepsilon_2$ and $T_1/T_2$ are invariant, we can apply generalized $\pi$-theorem (Sonin, 2004) and to re-write (17) and (18) as:

$$P_1 = T_1 \Psi_1 [V, \varepsilon_1/a] \tag{19}$$

$$P_2 = T_2 \Psi_2 [V, \varepsilon_2/a] \tag{20}$$

It can be seen that equations (19) and (20) are identical to the equation (7) and the response periods to obliquity and to precession do not depend on each other. This result is not by any means intuitive.

We now repeat the same reasoning for the corresponding amplitudes of the system response:

$$\acute{S}_1 = \varphi_1(V, a, \varepsilon_1, P_1, \varepsilon_2, P_2) \tag{21}$$

$$\acute{S}_2 = \varphi_2(V, a, \varepsilon_1, P_1, \varepsilon_2, P_2) \tag{22}$$

$$\acute{S}_1 = \varepsilon_1^2 P_1^2 \Phi_1(V, \varepsilon_1/a, \varepsilon_1/\varepsilon_2, P_1/P_2) \tag{23}$$

$$\acute{S}_2 = \varepsilon_2^2 P_2^2 \Phi_2(V, \varepsilon_2/a, \varepsilon_1/\varepsilon_2, P_1/P_2) \tag{24}$$

Though in the case of the orbital forcing $\varepsilon_1\varepsilon_2$ and $T_1/T_2$ are invariant, $P_1/P_2$ is not an invariant (see Fig. 1), therefore:

$$\acute{S}_1 = \varepsilon_1^2 P_1^2 \Phi_1(V, \varepsilon_1/a, P_1/P_2) \tag{25}$$

$$\acute{S}_2 = \varepsilon_2^2 P_2^2 \Phi_2(V, \varepsilon_2/a, P_1/P_2) \tag{26}$$

We can see that although periods of the system response to the precession and obliquity forcings are independent, the amplitudes of the corresponding variations are interdependent and thus may deviate from a pure square-period law. This observation may have an important implication for our understanding of the paleo data. As we demonstrated before (Verbitsky et al, 2018), $P_1/P_2$ evolves over time, specifically $P_1/P_2 = 1$ for the early Pleistocene due to precession period doubling and $P_1/P_2 = 4$ for the late Pleistocene due to obliquity period doubling. It means that the *slope* of the spectrum of the system response may also evolve.
    Introduction of more sinusoids (for example, accounting for the millennial forcing) makes the situation even more complex. In such a case, a period of the system response to a specific forcing component depends on the amplitudes and the periods of all sinusoids:

$$P = \psi[V, a, \varepsilon_1, T_1, \varepsilon_2, T_2, \dots \varepsilon_i, T_i \dots] \tag{27}$$

Then, for example, $P_1$, the period of the system response to obliquity forcing, can be presented as:

$$P_1 = T_1 \Psi_1 \left[ V, \frac{\varepsilon_1}{a}, \dots, \frac{\varepsilon_1}{\varepsilon_i}, \frac{T_1}{T_i}, \dots \right] \tag{28}$$

and corresponding amplitude of the glaciation area response

$$\acute{S}_1 = \varphi_1 [V, a, \varepsilon_1, P_1, \varepsilon_2, P_2, \dots \varepsilon_i, P_i \dots] \tag{29}$$

$$\acute{S}_1 = \varepsilon_1^2 P_1^2 \Phi_1 \left[ V, \frac{\varepsilon_1}{a}, \dots, \frac{\varepsilon_1}{\varepsilon_i}, \frac{P_1}{P_i}, \dots \right] \tag{30}$$

Equations (28) and (30) show that, generally speaking, every peak $P$ and corresponding amplitude $\acute{S}$ of the system response depend on each forcing sinusoid. Such dependence may break the scale invariance we discussed earlier. For example, we have demonstrated in our previous study (Verbitsky et al, 2019a) that introduction of the millennial variability of significant amplitude (i.e., $\varepsilon_1/\varepsilon_i \rightarrow 0$) may disrupt the system's response to the orbital forcing and essentially reduce the slope $\beta_a$. The empirical energy density spectrum of Huybers and Curry (2006) has the slope of B $\approx$ 2 in the orbital domain. Since the energy density slope B

relates to the fluctuation amplitude slope $\beta_a$ as $B = 2\beta_a + 1$, $B \approx 2$ corresponds to $\beta_a = 0.5 < 2$. We may therefore speculate that the observed spectrum of the climate variability could be significantly influenced by the millennial forcing propagated into the orbital domain.

### 4.3 How general is the property of scale invariance?

It is apparent that not every dynamical model has the property of scale invariance that is encoded in its dynamical equations. As an illustration, let us consider the van der Pol oscillator. It was previously suggested as a minimal model capturing ice-age dynamics (Crucifix, 2012):

$$\frac{dx}{dt} = \frac{-y+\beta+\gamma F}{\tau} \tag{31}$$

$$\frac{dy}{dt} = \frac{-\alpha(\frac{y^3}{3}-y-x)}{\tau} \tag{32}$$

Here all variables and parameters, except $\tau$, are dimensionless; $\tau$ is measured in units of time. Variable $x$ is thought to represent the global ice volume, and variable $y$ makes the "rest-of-the climate" response. Using the same $\pi$-theorem technique, let's determine the period $P$ and the amplitude $x'$ of the system response to the external forcing $F$ of the period $T$.

$$P = \psi(\alpha, \beta, \gamma, \tau, T) \tag{33}$$

$$P = T\Psi(\alpha, \beta, \gamma, \tau/T) \tag{34}$$

Since $\alpha$, $\beta$, and $\gamma$ are constants,

$$P = T\Psi(\tau/T) \tag{35}$$

Similarly,

$$x' = \varphi(\alpha, \beta, \gamma, \tau, P) \tag{36}$$

$$x' = \Phi(\alpha, \beta, \gamma, \tau/P) = \Phi(\tau/P) \tag{37}$$

It means that the amplitudes of forced fluctuations in the van der Pol model are not necessarily scale invariant. We have tested this conclusion experimentally for $\tau = 36.2$ kyr and a forcing period $T$ ranging from 5 kyr to 100 kyr. The response shows slope breaks near about 90-kyr and 50-kyr that are clearly related to the auto-oscillation of the 100-kyr dominant period and its 50-kyr over-tone.

Therefore, in a search for the most adequate ice-age physics, it would indeed be useful to see whether more sophisticated ice sheet – ocean – atmosphere models have the property of scale invariance. We suspect that potential universality of this property may stem from the universality of the equation (1). Equation (1) represents the global ice volume balance and simply says that changes of the ice volume are equal to the mass influx to the ice-sheet surface. This statement is valid for each and every climate model of any complexity. Therefore, if a model can be diagnosed with a single dimensionless number similar to the $V$-number that would effectively capture most of the climate dynamics, then the scale invariance of the glaciation area variations ($m^2$) can be reduced from the simple observation that it depends on the mass influx to its surface (m/s) and the periodicity of the mass influx variations (s). This might not be too difficult to verify with an adequate set of experiments, but we must obviously leave this task to the scientists who know and develop these models.

## 5. Conclusions.

Twenty-seven years ago, Saltzman and Verbitsky (1993) discussed their model of 18 parameters, 9 of which were physically unconstrained (i.e., free parameters) and formulated a challenge "to account for as much of the variance with fewer free parameters. A challenge is thus posed to ourselves and other theoretical paleoclimatologists to construct a more parsimonious model in this regard that can supersede our present effort." We may now conclude that this challenge has been met. All parameters in our model (Verbitsky et al, 2018) are physically constrained. Moreover, dimensional analysis reveals that *only two factors define most of the ice-age dynamics*: (a) a balance between intensities of climate positive and ice sheet negative feedbacks, $\Pi_1 = V$; and (b) the period, $T$, and the amplitude of the external forcing, $\varepsilon$, (specifically, a particular proportion between the external, e.g., orbital, and terrestrial ice sheet mass balance components, $\Pi_2 = \varepsilon/a$).

The analysis indicates that the amplitudes of glacial area variations and of climate temperature are *scale invariant* with a frequency slope of 2. The property of scale invariance does not depend on the physical nature of the underlying positive and negative feedbacks incorporated by the system. It thus turns out to be one of the most fundamental properties of the Pleistocene climate.

Retrospectively, we could have inferred scale invariance from the mere assumption that the behavior of the continental glacial area (measured in $m^2$) depends on the mass influx to its surface (m/s) and the periodicity of the mass influx variations (s), but perhaps these assumptions are too simple to be convincing. In our study, we have chosen a bit more sophisticated but more credible approach. We derived a dynamical model from the scaled conservation equations of viscous non-Newtonian ice combined with an equation describing the evolution of the climate temperature. We observed that most of the dynamical system behavior can be explained by a balance between positive and negative feedbacks. This observation, finally, illuminated the crucial role of the mass influx and its periodicity, making application of the π-theorem effective and definitive.

Certainly, we cannot claim to have a full picture of the mechanisms of ice ages, but if ice age physics are well captured by the mathematical structure that we have obtained, then this scale invariance linking response amplitudes and periods applies. We further suggest that a model that would indeed be a bit different than the Verbitsky et al (2018) model because it includes some other important (may be non-liner) mechanisms, might still retain an important property that we have discovered: there is a connection between the sensitivity of the fixed point (since the *V*-number is indeed constructed by consideration of the sensitivity of the fixed point) and a scale invariance linking period and amplitude of response. This seems to be the fundamental proposal, for which we welcome challengers equipped with bigger models.

**Code and data availability.** The MatLab R2015b code and data to calculate model response to periodical forcing as it is presented in Fig.2 (Verbitsky et al, 2019b) are available at http://doi.org/10.5281/zenodo.3473957 (last access: October 5, 2019)

**Author contributions:** MYV conceived the research and developed the formalism. MYV and MC contributed equally to writing the manuscript.

**Competing interests:** The authors declare that they have no conflict of interest.

**Acknowledgements:** We are grateful to our three anonymous reviewers for their helpful comments and to Dmitry Volobuev for his help in producing Figure 2.

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
