# Peer review of "Π*-theorem generalization of the ice-age theory"

_Earth System Dynamics, 2019_

## Referee Comment (RC1) · Anonymous Referee #1 · 30 Nov 2019

This paper uses an idealized model of ice age and the Buckingham-pi dimensional analysis theorem to gain insight into glacial cycles. The approach is interesting and can potentially benefit the community. I have a few comments and suggestions that I ask the authors to address.

Specific comments - I am a bit confused about why the authors have not gone beyond deriving equations such as (9) or (13) to actually find the full scaling relationship, as is often done (e.g., see the papers I mentioned in my very last comment). What I mean is to find the functional form of $\phi$ or $X$ in these equations by computing the powers of $\Pi_1$ and $\Pi_2$ in Eq. (9) and $\Pi_1 - \Pi_4$ in Eq (13) using simulations. Even if the whole goal is to find scale invariances, then this is important: in the analysis of Eq. (13), the authors state that because $\Pi_3$ and $\Pi_4$ include $P$, then $\theta'$ is not expected to be scale

invariant. But it is possible that if you find the functional form of $X$, you find something like $\theta' \sim P \times \ldots \Pi_3^A \times \Pi_4^B$ with $A = -B$. In that case, $P$ drops out from $X$ and $\theta'$ would be scale invariant with $P$. The authors should do this analysis, or fully explain why it is not needed, and also address the issue I raised about their interpretation of Eq. (13).

- While there is great value in idealized models, and as the authors clearly stated, the dimensional analysis could be only effectively applied to an idealized model, I believe that the authors should at the end, test, or discuss the implications of, their findings in the context of data from more comprehensive models or actual observations (proxies). That would really demonstrate the power of this approach and increase the impact of this work.

Minor comments/suggestions

- Line 47: explicitly mention that in this case, one gets 18-4=14 pi groups

- It is up to the authors, but I suggest using the word "dimensionless" instead of "adimensional"

- Line 45: what is the unit of concentration in terms of fundamental dimensions?

- It is up to the authors, but I suggest using Kelvin (K) instead of degree Celsius (C) as the unit of temperature

- Fig 1: improve the clarity of the figure and expand the caption. Also, What is the line with $\beta_a = 1$?

- Lines 39-41: There are a few papers in which the Buckingham-pi theorem is applied to a problem in global climate dynamics or its low-dimensional model,

MJO: Yang, D. and Ingersoll, A.P., 2014. A theory of the MJO horizontal scale. Geophysical Research Letters, 41(3), pp.1059-1064.

Planetary circulation: Koll, D.D. and Abbot, D.S., 2015. Deciphering thermal phase curves of dry, tidally locked terrestrial planets. The Astrophysical Journal, 802(1), p.21.

Blocking events: Nabizadeh, E., Hassanzadeh, P., Yang, D. and Barnes, E.A., 2019. Size of the atmospheric blocking events: Scaling law and response to climate change. Geophysical Research Letters. 46

---

## Referee Comment (RC2) · Anonymous Referee #2 · 4 Dec 2019

As ice-age dynamics is not my field of research, I have been kindly asked by the Editor to verify whether the math is correct and whether the application of the Buckingham-Pi theorem is adequate to this system. So from that perspective I haven't found any mistake and I find the application of the Pi theorem to be very nice.

Furthermore, the way the authors defined the non-dimensional V number as the ratio between the positive to negative feedback mechanisms is illuminating (although they have done it in a previous paper - Verbitsky et al., 2018, so this by itself is not a new result of the current paper). It is also quite impressing that, at least for a simple type of forcing, the overall response of the dynamical system depends to a high degree on the value of V rather than on the details of the different combinations of positive and negative feedback mechanisms. The authors themselves are honest to show the

limitations of this approach for a more complex type of forcing like a multi sinusoidal forcing. So in terms of what is new in this paper - I think the authors managed now to incorporate their previous works into a more coherent mathematical framework based on, first - a drastic reduction of the number of parameters by defining the V number, and second by a further reduction using the Pi similarity theorem.

Therefore overall I think the paper deserves publication. I personally myself feel a bit uncomfortable with the starting point of a model of the type of eqs' (1-3). In one hand it is a low order model but on the other hand it is still quite complex. When I see such models I always get a feeling that maybe there are other equally important feedback mechanisms that are not included and maybe they will change dramatically the dynamical behavior.

Nonetheless, the robustness of the V number, at least when subject to a simple forcing, is impressive. Therefore, in order to strengthen the paper, and add more new material, my suggestion is that the authors will take this model and add several potential feedback mechanisms to obtain different variation of dynamical systems. Then they will have different V numbers for the different models. If for the same value of the different V numbers, for the different models, the dynamic response of the different models will be similar this will be highly cool and much more robust. This will mean that what truly matters is the ratio between positive to negative feedback mechanisms, not only within the same model but also with similar models of the same family. I will be happy to review the revised version.

---

## Author Comment (AC1) · 16 Dec 2019

Dear Anonymous Referee #1,

Thank you for your detailed review and insightful suggestions. We are pleased to learn that you find our approach to be interesting and helpful. The following is our response to your comments and suggestions.

Comment: I am a bit confused about why the authors have not gone beyond deriving equations such as (9) or (13) to actually find the full scaling relationship, as is often done (e.g., see the papers I mentioned in my very last comment). What I mean is to find the functional form of $\varphi$ or X in these equations by computing the powers of $\Pi 1$ and $\Pi 2$ in Eq. (9) and $\Pi 1 - \Pi 4$ in Eq (13) using simulations. Even if the whole goal is

to find scale invariances, then this is important: in the analysis of Eq. (13), the authors state that because $\Pi3$ and $\Pi4$ include P, then $\theta$ is not expected to be scale invariant. But it is possible that if you find the functional form of X, you find something like $\theta$' âĹij P×. . . ..$(\Pi3)\hat{}A \times (\Pi4)\hat{}B$ with A = $-$B. In that case, P drops out from X and $\theta$' would be scale invariant with P. The authors should do this analysis, or fully explain why it is not needed, and also address the issue I raised about their interpretation of Eq. (13).

Answer: As we discuss in the paragraph 4.1, the property of the scale invariance does not depend on the physical nature of the underlying positive and negative feedbacks that define the V-number ($\Pi1$). At the same time, the function $\Phi$ of the equation (9) and the function X of the equation (13) do depend on the underlying physics. To calculate functions $\Phi$ or X as powers of $\Pi1$ and $\Pi2$ in Eq. (9) and $\Pi1 - \Pi4$ in Eq (13), we would need to span the space of eight (8) parameters forming the V-number. Obviously, this would defeat the purpose of this research, and therefore, we have limited ourselves with the discovery of the scale invariance only. Nevertheless, your observation regarding the equation (13) is correct. We have also observed experimentally that when the amplitude of the external forcing, $\varepsilon$, is reduced, the equation (13) becomes scale invariant with a frequency slope equal 1. In this case $\theta$' $\sim$ X($\Pi1$, $\Pi2$, $\Pi3/\Pi4$). We did not include this analysis into the paper because the effect of a reduction of the amplitude of the astronomical forcing is not something we expect to see in the real world. However, in retrospect and given your comment, we see the benefit of bringing this analysis back, because it will hopefully make our thinking more explicit.

Action: We will add this discussion into the text

Comment: While there is great value in idealized models, and as the authors clearly stated, the dimensional analysis could be only effectively applied to an idealized model, I believe that the authors should at the end, test, or discuss the implications of, their findings in the context of data from more comprehensive models or actual observations (proxies). That would really demonstrate the power of this approach and increase the impact of this work.

Answer: This is a good point, but difficult to address in practice. We would like to take this opportunity to share our views about how our study, we believe, contributes to filling a gap in the literature. Palaeoclimate simulations with "more sophisticated models", including the seminal paper by Abe-Ouchi et al., 2013, and the simulations with CLIMBER provided by Ganopolski et al. 2010, tend to focus on the response of the ice-sheet climate system to orbital forcing, and discuss the respective amplitudes of the 100-ka, 41-ka, and 21-23-ka periods, but none discuss the slope of the power spectrum down to the millennium scale. Yet, empirical analysis of paleoclimate series shows that there is a rich spectral content and point to the existence of "spectral slopes" (to cite by a few, Huybers and Curry 2006 and Lovejoy and Schertzer, 2013). Lovejoy and Schertzer evoke some generic process, such as the principle of "cascades" and which is tightly linked to the concept of scale invariance of the equations. For example, the scale invariance of fluid-dynamics equations is exploited to provide inferences about spectral slopes of turbulent flows. However, to our knowledge, there is no available theory supporting scale invariance in regimes associated with glacial-interglacial dynamics. So, we believe that we have here been providing at least some important elements that should help us to bridge both approaches. If the sensitivity of the stationary state is effectively determined by a dimensionless number (the V-number) in the way our model does, then we satisfy a necessary condition to produce relationship between the amplitude and duration of glacial cycles over a reasonably wide range of periods, including the millennial scales. It would indeed be useful to see whether a similar response-scaling structure appears with more sophisticated ice-sheet-atmosphere model. This might not be too difficult to verify with an adequate set of experiments, but we must obviously leave this task to the scientists who know and develop these models. Perhaps, though, it is worth restating the physical roots of our enterprise. Our model was developed with attention to scaling invariance of ice flow conservation laws (Verbitsky et al. 2018), and was also tested against the ice-sheet-ice-shelf model of Pollard and De Conto (2012). Of course, we fully appreciate that there is some mileage left before delivering of a full theory of the fluctuation spectrum, from millennial to glacial-interglacial cycles. This objective, among others, requires understanding better the structure of the millennial variability, which was here merely postulated as a forcing. Hopefully the reviewer will understand that need to proceed step by step.

Action: We will add this discussion into the text

Minor comments/suggestions

Line 47: explicitly mention that in this case, one gets 18-4=14 pi groups; Action: Will be done

It is up to the authors, but I suggest using the word "dimensionless" instead of "adimensional" Action: Will be done

Line 45: what is the unit of concentration in terms of fundamental dimensions? It is up to the authors, but I suggest using Kelvin (K) instead of degree Celsius (C) as the unit of temperature Answer: $CO_2$ concentration is usually measured in ppm, parts per million, or mg/L. Since we mention these units in a reference to a specific model and its variables (Saltzman and Verbitsky, 1993), we think we need to keep the units of measurements that the authors used in their model.

Fig 1: improve the clarity of the figure and expand the caption. Also, what is the line with $\beta a = 1$? Action: Higher quality pictures (including better captions) will be provided

Lines 39-41: There are a few papers in which the Buckingham-pi theorem is applied to a problem in global climate dynamics or its low-dimensional model, MJO: Yang, D. and Ingersoll, A.P., 2014. A theory of the MJO horizontal scale. Geophysical Research Letters, 41(3), pp.1059-1064. Planetary circulation: Koll, D.D. and Abbot, D.S., 2015. Deciphering thermal phase curves of dry, tidally locked terrestrial planets. The Astrophysical Journal, 802(1), p.21. C2 ESDD Interactive comment Printer-friendly version Discussion paper Blocking events: Nabizadeh, E., Hassanzadeh, P., Yang, D. and Barnes, E.A., 2019. Size of the atmospheric blocking events: Scaling law and response to climate change. Geophysical Research Letters. 46

[Figure]

Answer: We agree. Indeed, if we say "low-order models of global climate dynamics" we should mention the references you provided. Otherwise, we need to narrow our statement, like, for example "low-order models of the Pleistocene climate" Action: The sentence will be edited.

References:

Abe-Ouchi A., F. Saito, K. Kawamura, M. E. Raymo, J. I. Okuno, K. Takahashi and H. Blatter (2013), Insolation-driven 100,000-year glacial cycles and hysteresis of ice-sheet volume, Nature, (500) 190–193 doi:10.1038/nature12374

Ganopolski A., R. Calov and M. Claussen (2010), Simulation of the last glacial cycle with a coupled climate ice-sheet model of intermediate complexity, Climate of the Past, (6) 229–244 doi:10.5194/cp-6-229-2010

Huybers P. and W. Curry (2006), Links between annual, Milankovitch and continuum temperature variability, Nature, (441) 329-332 doi:10.1038/nature04745

Lovejoy, Shaun, and Daniel Schertzer. The weather and climate: emergent laws and multifractal cascades. Cambridge University Press, 2013.

Pollard D. and R. M. DeConto (2012), Description of a hybrid ice sheet-shelf model, and application to Antarctica, Geoscientific Model Development, (5) 1273–1295 doi:10.5194/gmd-5-1273-2012

Verbitsky M. Y., M. Crucifix and D. M. Volobuev (2018), A theory of Pleistocene glacial rhythmicity, Earth System Dynamics, (9) 1025–1043 doi:10.5194/esd-9-1025-2018

---

## Referee Comment (RC3) · Anonymous Referee #3 · 17 Dec 2019

This is an interesting contribution to our understanding of the ice ages and the structure of glacial cycles. However, a broader review of ice age dynamics is needed in the introduction and in the wider paper. This will make it more accessible to a wider audience of Quaternary scientists. For the introduction, some reference to studies of ice age dynamics would be useful. There are obviously lots of paper you can refer to here, such as Imbrie et al. 1993, Paillard 2001 and Lang and Wolff 2011, for example.

In particular, it would be useful if you can explain more clearly and explicitly the wider significance of your findings for understanding the nature of glacial cycles. Your paper is clearly important because it provides a mathematical solution for understanding ice-age dynamics whereas other approaches are more qualitative or semi-quantitative (e.g. Hughes and Gibbard, 2018). However, Hughes and Gibbard (2018) found that our

understanding of glacial cycles, especially ice dynamics, is not always easily explained by external forcing such as solar radiation, although this does account for ~50-60% of glacier change and associated sea level change through glacial cycles. Internal glacier-climate dynamics account for the rest of the glacier variations. A complex interplay of various geographical factors was found to be responsible for the asynchronous spatial variation in global glacier dynamics, in both the largest high- and mid-latitude ice sheets as well as in smaller mountain ice caps and glaciers at a range of latitudes around the world. Your modelling appears to incorporate ice sheet dynamics only, and the feedbacks associated with this, and does not account for the complexity of the known spatial and temporal glacial patterns. Of course, I don't expect you to solve this in your modelling, but you should make the reader know that you aware of the limitations of your approach.

You conclude that only two factors define most of the ice age dynamics: a) a balance between intensities of climate positive and ice sheet negative dynamics and b) the period T and the amplitude of the external forcing. I can see how for b) this can be constrained from orbital parameters but the variables for a) are potentially very complex and only partially accounted for in your modelling. From this, if we can be confident about b) it would be useful to see a statement on the comparable effects of a) versus b). You may already do this, but I would like to see a much clearer statement on this matter. For example, be much clearer about the implications of what you mean by "the amplitude and duration of glacial cycles is governed by a property of scale-invariance that does not depend on the underlying positive and negative feedbacks incorporated by the system". Unless you make the wider significance your findings more explicit, then it will have a limited audience. I think the findings are potentially very important, and you need to communicate these more effectively with those researching ice age dynamics, who will then be able to refer to your work, thereby increasing the academic impact of this paper.

Refs:

Hughes, P.D., Gibbard, P.L., 2018. Global glacier dynamics during 100 ka Pleistocene glacial cycles. Quaternary Research 90, 222-243.

Imbrie, J., Hays, J.D., Martinson, D.G., McIntyre, A., Mix, A.C., Morley, J.J., Pisias, N.G., Prell, W.L., Shackleton, N.J., 1984. The orbital theory of Pleistocene climate: support from a revised chronology of the marine 18O record. In: Berger, A., Imbrie, J., Hays, G., Kukla, G., Saltzman, B. (Eds.), Milankovitch and Climate. Reidel, Dortrecht, p. 269–306.

Lang, N., Wolff, E.W., 2011. Interglacial and glacial variability from the last 800 ka in marine, ice and terrestrial archives. Climate of the Past 7, 361-380. doi:10.5194/cp-7-361-2011

Paillard, D., 2001. Glacial cycles: Towards a new paradigm. Reviews of Geophysics 39, 325-346.

---

## Author Comment (AC4) · 24 Dec 2019

Dear Anonymous Referee #3,

Thank you very much for your detailed review and helpful suggestions. We appreciate that you consider our findings to be important. The following is our response to your suggestions.

Suggestion: This is an interesting contribution to our understanding of the ice ages and the structure of glacial cycles. However, a broader review of ice age dynamics is needed in the introduction and in the wider paper. This will make it more accessible to a wider audience of Quaternary scientists. For the introduction, some reference to studies of ice age dynamics would be useful. There are obviously lots of papers you

can refer to here, such as Imbrie et al. 1993, Paillard 2001 and Lang and Wolff 2011, for example.

Answer: We agree that the introduction may be expanded. Currently, it is focused on the unique properties of our model that allows us to use the $\pi$-theorem insightfully. At the same time, it does not provide enough background that would allow our readers to better appreciate the importance of the obtained results.

Action: We will expand the introduction with a brief review of the state-of-the-art ice age research.

Suggestion: In particular, it would be useful if you can explain more clearly and explicitly the wider significance of your findings for understanding the nature of glacial cycles. Your paper is clearly important because it provides a mathematical solution for understanding ice age dynamics whereas other approaches are more qualitative or semi-quantitative (e.g. Hughes and Gibbard, 2018). However, Hughes and Gibbard (2018) found that our understanding of glacial cycles, especially ice dynamics, is not always easily explained by external forcing such as solar radiation, although this does account for 50-60% of glacier change and associated sea level change through glacial cycles. Internal glacier climate dynamics account for the rest of the glacier variations. A complex interplay of various geographical factors was found to be responsible for the asynchronous spatial variation in global glacier dynamics, in both the largest high- and mid-latitude ice sheets as well as in smaller mountain ice caps and glaciers at a range of latitudes around the world. Your modelling appears to incorporate ice sheet dynamics only, and the feedbacks associated with this, and does not account for the complexity of the known spatial and temporal glacial patterns. Of course, I don't expect you to solve this in your modelling, but you should make the reader know that you aware of the limitations of your approach.

Answer: Your observation, that the ice-sheet dynamics (equations (1) and (2)) is the most comprehensive and most physically substantiated part of the model, is cor-

rect. The equation (3) of the "rest-of-the-climate" is ambiguous but its ambiguity allows us to interpret our experiments with increased (or reduced) $\gamma 2$ as experiments with additional feedbacks of different nature (see also our response to Anonymous Referee #2 https://www.earth-syst-dynam-discuss.net/esd-2019-65/esd-2019-65-AC2-supplement.pdf). In other words, we are uncertain about some key mechanisms that we have chosen to describe using the "rest-of-the-climate" linear equation. Among others, non-linear effects related to the carbon cycle, non-linear effects related to sea-level destabilization of ice sheets and related synchronization, non-linear effects related to atmospheric circulation, or non-linear effects related to biogenic calcifiers and their action on alkalinity, etc. A challenger might thus claim that these effects are so important that they should be taken off $\gamma 2$ and be considered more explicitly. However, we have the hope that even after accounting for these processes, we might end up with a model that still has grossly the same mathematical structure as the Verbitsky et al (2018) model, even though the meaning of some of the variables will have changed.

Action: We will add the above discussion into the text.

Suggestion: You conclude that only two factors define most of the ice age dynamics: a) a balance between intensities of climate positive and ice sheet negative dynamics and b) the period T and the amplitude of the external forcing. I can see how for b) this can be constrained from orbital parameters but the variables for a) are potentially very complex and only partially accounted for in your modelling. From this, if we can be confident about b) it would be useful to see a statement on the comparable effects of a) versus b). You may already do this, but I would like to see a much clearer statement on this matter. For example, be much clearer about the implications of what you mean by "the amplitude and duration of glacial cycles is governed by a property of scale-invariance that does not depend on the underlying positive and negative feedbacks incorporated by the system". Unless you make the wider significance your findings more explicit, then it will have a limited audience. I think the findings are potentially very important, and you need to communicate these more effectively with those researching ice age

dynamics, who will then be able to refer to your work, thereby increasing the academic impact of this paper.

Answer: Unlike many models of the ice-age climate that postulate internal 100-kyr oscillator, in our model, 100-kyr cycle is produced as a non-linear system response to the astronomical forcing. Nothing happens without astronomical forcing and nothing happens without system internal dynamics. Therefore, it is not possible to quantify precisely the impact of the astronomical forcing versus internal climate dynamics. Furthermore, a similar system response may be observed with different forcing and different internal climate dynamics. For illustration, let us consider equation (7)

$P=T\Psi(V,\varepsilon/a)$

and equation (9)

$S'=\varepsilon\text{\textasciicircum}2\ P\text{\textasciicircum}2\ \Phi(V,\varepsilon/a)$

Here the V-number is defined by the climate dynamics, $\varepsilon/a$ is the relative intensity of the astronomical forcing, T is the forcing period, P is the period of system response, and S' is the amplitude of the system response. Function $\Psi(V,\varepsilon/a)$ defines a forcing-period doubling domain and it may be the same (let say, $\Psi=2$) for different combinations of V and $\varepsilon/a$ (see also Fig. 1). We can only say that the obliquity-period doubling requires both well-developed positive feedbacks in the system ($0.6 < V < 0.8$) and relatively high climate sensitivity to the astronomical forcing ($\varepsilon/a \approx 1$). Moreover, different sets of parameters may lead to the same V-number. Function $\Phi(V,\varepsilon/a)$ and function $\Phi'(V,\varepsilon/a)$ corresponding to the same value of the V-number but formed by the different parameters may not be the same. Similarly, function $\Psi(V,\varepsilon/a)$ and function $\Psi'(V,\varepsilon/a)$ may differ. Most remarkably though, the power degree "2" of the response-period in the equation (9) is defined by the fundamental dimensionality requirements and does not depend on the underlying physics. This is what gives us the property of scale invariance but at the same time makes our efforts to disambiguate historical records even more challenging.

As a result, we can't claim to have a full picture of the mechanisms of ice ages, but if ice age physics are well captured by the mathematical structure that we have obtained, then this scale invariance linking amplitude and response periods applies. We further suggest that a model that would indeed be a bit different than the Verbitsky et al (2018) model because it includes some other important (may be non-liner) mechanisms, might still retain an important property that we have discovered: there is a connection between the sensitivity of the fixed point (since the V-number is indeed constructed by consideration to the sensitivity of the fixed point) and a scale invariance linking period and amplitude of response. This seems to be the fundamental proposal, for which we welcome challengers equipped with bigger models.

Action: We will add the above discussion into the text.

References:

Verbitsky M. Y., M. Crucifix and D. M. Volobuev (2018), A theory of Pleistocene glacial rhythmicity, Earth System Dynamics, (9) 1025–1043 doi:10.5194/esd-9-1025-2018

---

## Author Response (AR1)

Dear Dr. Messori,

Thank you for your decision. The interactive discussion process has been very useful and brought quite a few good suggestions. You will see in the point-by-point reply below and in the marked-up manuscript that we addressed all of them. Indeed, we also responded to the comment of the Reviewer 1, that you quote "While there is great value in idealized models, and as the authors clearly stated, the dimensional analysis could be only effectively applied to an idealized model, I believe that the authors should at the end, test, *or discuss* the implications of, their findings in the context of data from more comprehensive models or actual observations (proxies). That would really demonstrate the power of this approach and increase the impact of this work".

Though actual data from three-dimensional ice sheet – ocean - atmosphere models are not available to us for a number of reasons that we outline in our extended response to Reviewer 1 (see pp. 2-3 below), we, to a great extent, discuss the implications of our findings and formulate the challenges for the scientists who own and develop such models. Specifically,

(1) In the paragraph 4.1 we added an analysis to demonstrate that our climate equation (3) may represent a number of feedbacks of different nature;

(2) We made a new paragraph 4.3 "How general is the property of scale invariance?" and propose that potential universality of scale invariance may stem from the universality of the equation (1) that represents the balance of global ice volume and is valid for each and every climate model of any complexity;

(3) We added additional discussion to the Introduction and Conclusions sections

Please note that our results have already been discussed in the paragraph 4.2 in the context of the empirical power spectrum of Huybers and Curry (2006).

Dr. Messori, We believe that with this paper we are on a groundbreaking territory, because so far there is no available theory supporting scale invariance in regimes associated with glacial-interglacial dynamics. At the same time, we, indeed, fully recognize that a lot still needs to be done and consider our paper only as a first step in the process of developing a full theory of the fluctuation spectrum, from orbital to sub-orbital (millennial) cycles.

January 11, 2020

**Response to Anonymous Referee #1**

Dear Anonymous Referee #1, Thank you for your detailed review and insightful suggestions. We are pleased to learn that you find our approach to be interesting and helpful. The following is our response to your comments and suggestions.

**Comment:** I am a bit confused about why the authors have not gone beyond deriving equations such as (9) or (13) to actually find the full scaling relationship, as is often done (e.g., see the papers I mentioned in my very last comment). What I mean is to find the functional form of φ or X in these equations by computing the powers of $\Pi_1$ and $\Pi_2$ in Eq. (9) and $\Pi_1 - \Pi_4$ in Eq (13) using simulations. Even if the whole goal is to find scale invariances, then this is important: in the analysis of Eq. (13), the authors state that because $\Pi_3$ and $\Pi_4$ include P, then θ is not expected to be scale invariant. But it is possible that if you find the functional form of X, you find something like $\theta' \sim P \times \ldots \ldots (\Pi_3)^{\wedge}A \times (\Pi_4)^{\wedge}B$ with A = −B. In that case, P drops out from X and θ' would be scale invariant with P. The authors should do this analysis, or fully explain why it is not needed, and also address the issue I raised about their interpretation of Eq. (13).

**Answer:** As we discuss in the paragraph 4.1, the property of the scale invariance does not depend on the physical nature of the underlying positive and negative feedbacks that define the V-number ($\Pi_1$). At the same time, the function Φ of the equation (9) and the function X of the equation (13) do depend on the underlying physics. To calculate functions Φ or X as powers of $\Pi_1$ and $\Pi_2$ in Eq. (9) and $\Pi_1 - \Pi_4$ in Eq (13), we would need to span the space of eight (8) parameters forming the V-number. Obviously, this would defeat the purpose of this research, and therefore, we have limited ourselves with the discovery of the scale invariance only.

Nevertheless, your observation regarding the equation (13) is correct. We have also observed experimentally that when the amplitude of the external forcing, ε, is reduced, the equation (13) becomes scale invariant with a frequency slope equal 1. In this case $\theta' \sim X(\Pi_1, \Pi_2, \Pi_3/\Pi_4)$. We did not include this analysis into the paper because the effect of a reduction of the amplitude of the astronomical forcing is not something we expect to see in the real world. However, in retrospect and given your comment, we see the benefit of bringing this analysis back, because it will hopefully make our thinking more explicit.

**Action:** We will add this discussion into the text. **Done: p.13, lines 15-18**

**Comment:** While there is great value in idealized models, and as the authors clearly stated, the dimensional analysis could be only effectively applied to an idealized model, I believe that the authors should at the end, test, or discuss the implications of, their findings in the context of data from more comprehensive models or actual observations (proxies). That would really demonstrate the power of this approach and increase the impact of this work.

**Answer:** This is a good point, but difficult to address in practice. We would like to take this opportunity to share our views about how our study, we believe, contributes to filling a gap in the literature.

Palaeoclimate simulations with "more sophisticated models", including the seminal paper by Abe-Ouchi et al., 2013, and the simulations with CLIMBER provided by Ganopolski et al. 2010, tend to focus on the response of the ice-sheet climate system to orbital forcing, and discuss the respective amplitudes of the 100-ka, 41-ka, and 21-23-ka periods, but none discuss the slope of the power spectrum down to the millennium scale.

Yet, empirical analysis of paleoclimate series shows that there is a rich spectral content and point to the existence of "spectral slopes" (to cite by a few, Huybers and Curry 2006 and Lovejoy and Schertzer, 2013). Lovejoy and Schertzer evoke some generic process, such as the principle of "cascades" and which is tightly linked to the concept of scale invariance of the equations. For

example, the scale invariance of fluid-dynamics equations is exploited to provide inferences about spectral slopes of turbulent flows. However, to our knowledge, there is no available theory supporting scale invariance in regimes associated with glacial-interglacial dynamics.

So, we believe that we have here been providing at least some important elements that should help us to bridge both approaches. If the sensitivity of the stationary state is effectively determined by a dimensionless number (the V-number) in the way our model does, then we satisfy a necessary condition to produce relationship between the amplitude and duration of glacial cycles over a reasonably wide range of periods, including the millennial scales. It would indeed be useful to see whether a similar response-scaling structure appears with more sophisticated ice-sheet-atmosphere model. This might not be too difficult to verify with an adequate set of experiments, but we must obviously leave this task to the scientists who know and develop these models. Perhaps, though, it is worth restating the physical roots of our enterprise. Our model was developed with attention to scaling invariance of ice flow conservation laws (Verbitsky et al. 2018), and was also tested against the ice-sheet-ice-shelf model of Pollard and De Conto (2012).

Of course, we fully appreciate that there is some mileage left before delivering of a full theory of the fluctuation spectrum, from millennial to glacial-interglacial cycles. This objective, among others, requires understanding better the structure of the millennial variability, which was here merely postulated as a forcing. Hopefully the reviewer will understand that need to proceed step by step.

**Action:** We will add this discussion into the text. **Done: p.9 lines 17-30, p.17, lines 5-38; p.18, lines 22-29**

**Minor comments/suggestions**

**Line 47:** explicitly mention that in this case, one gets 18-4=14 pi groups; **Action: Done: p.8 lines 49-50, p.9 lines 4-5**

**It is up to the authors, but I suggest** using the word "dimensionless" instead of "adimensional" **Action: Done: p.8 line 17, p.14 line 5**

**Line 45:** what is the unit of concentration in terms of fundamental dimensions? It is up to the authors, but I suggest using Kelvin (K) instead of degree Celsius (C) as the unit of temperature **Answer:** $CO_2$ concentration is usually measured in ppm, parts per million, or mg/L. Since we mention these units in a reference to a specific model and its variables (Saltzman and Verbitsky, 1993), we think we need to keep the units of measurements that the authors used in their model.

**Fig 2:** improve the clarity of the figure and expand the caption. Also, what is the line with $\beta a = 1$? **Action:** Higher quality pictures (including better captions) will be provided. **Done, p. 12 Fig. 2**

**Lines 39-41:** There are a few papers in which the Buckingham-pi theorem is applied to a problem in global climate dynamics or its low-dimensional model, MJO: Yang, D. and Ingersoll, A.P., 2014. A theory of the MJO horizontal scale. Geophysical Research Letters, 41(3), pp.1059-1064. Planetary circulation: Koll, D.D. and Abbot, D.S., 2015. Deciphering thermal phase curves of dry, tidally locked terrestrial planets. The Astrophysical Journal, 802(1), p.21. C2 ESDD Interactive comment Printer-friendly version Discussion paper Blocking events: Nabizadeh, E., Hassanzadeh, P., Yang, D. and Barnes, E.A., 2019. Size of the atmospheric blocking events: Scaling law and response to climate change. Geophysical Research Letters. 46

**Answer:** We agree. Indeed, if we say "low-order models of global climate dynamics" we should mention the references you provided. Otherwise, we need to narrow our statement, like, for example "low-order models of the Pleistocene climate"
**Action:** The sentence will be edited. **Done: p.8 lines 39-40**

**Response to Anonymous Referee #2**
Dear Anonymous Referee #2, Thank you very much for your thorough review and a very interesting suggestion. We are delighted to hear that you find our research to be illuminating and impressive. The following is our response to your suggestion.

**Suggestion:** Therefore overall I think the paper deserves publication. I personally myself feel a bit uncomfortable with the starting point of a model of the type of eqs' (1-3). In one hand it is a low order model but on the other hand it is still quite complex. When I see such models I always get a feeling that maybe there are other equally important feedback mechanisms that are not included and maybe they will change dramatically the dynamical behavior.
Nonetheless, the robustness of the V number, at least when subject to a simple forcing, is impressive. Therefore, in order to strengthen the paper, and add more new material, my suggestion is that the authors will take this model and add several potential feedback mechanisms to obtain different variation of dynamical systems. Then they will have different V numbers for the different models. If for the same value of the different V numbers, for the different models, the dynamic response of the different models will be similar this will be highly cool and much more robust. This will mean that what truly matters is the ratio between positive to negative feedback mechanisms, not only within the same model but also with similar models of the same family. I will be happy to review the revised version.
**Answer:** The Verbitsky et al (2018) model is, to our knowledge, unique because it is the only low-order ice-age model that, instead of being postulated, has been parsimoniously reduced from the conservation equations of viscous ice flow. Equations (1) and (2) have been derived from the ice mass balance and the ice-flow energy equations, correspondingly. For this reason, we would prefer to keep them untouched. The equation (3) of the "rest-of-the-climate temperature" is, indeed, ambiguous, but since it is linear, it can be split into several equations:

$$\omega = \omega_1 + \omega_2 + \cdots + \omega_n$$

$$\frac{d\omega_1}{dt} = \gamma_{11} - \gamma_{21}(S - S_0) - \gamma_3\omega_1$$

$$\frac{d\omega_2}{dt} = \gamma_{12} - \gamma_{22}(S - S_0) - \gamma_3\omega_2$$
…
$$\frac{d\omega_n}{dt} = \gamma_{1n} - \gamma_{2n}(S - S_0) - \gamma_3\omega_n$$

Each of the above equations may represent different feedback mechanisms. Therefore our experiments with increased (or reduced) $\gamma_2$ may be also understood as experiments with additional feedbacks of different nature ($\gamma_2 = \gamma_{21} + \gamma_{22} + \ldots + \gamma_{2n}$), though of the same time-scale $1/\gamma_3$.
Certainly, if we introduce in our model more dramatic - although not necessarily more realistic - changes, the dynamics of the system may be different. As an illustration, let us consider the van

der Pol oscillator. It was previously suggested as a minimal model capturing ice-age dynamics (Crucifix, 2012):

$$\frac{dx}{dt} = \frac{-y + \beta + \gamma F}{\tau}$$

$$\frac{dy}{dt} = \frac{-\alpha(\frac{y^3}{3} - y - x)}{\tau}$$

Here all variables and parameters, except $\tau$, are dimensionless; $\tau$ is measured in units of time. Variable x is thought to represent the global ice volume, and variable y makes the "rest-of-the climate" response. Using the same $\pi$-theorem technique, let's determine the period P and the amplitude x' of the system response to the external forcing F of the period T.

$$P = \psi(\alpha, \beta, \gamma, \tau, T)$$

$$P = T\Psi(\alpha, \beta, \gamma, \tau / T)$$

Since $\alpha$, $\beta$, and $\gamma$ are constants

$$P = T\Psi(\tau / T)$$

Similarly,

$$x' = \varphi(\alpha, \beta, \gamma, \tau, P)$$

$$x' = \Phi(\alpha, \beta, \gamma, \tau/P) = \Phi(\tau/P)$$

It means that the amplitudes of forced fluctuations in the van der Pol model are not expected to be scale invariant. We have tested this conclusion experimentally for $\tau$ = 36.2 kyr (this reference value of $\tau$ produces auto-oscillations with a 100-kyr period) and a forcing period T ranging from 5 kyr to 100 kyr. Therefore, instead of comparing our model with other existing low-order models or creating a new low-order model for the sole purpose of a comparison, we think it would be more advantageous, in the future work, to compare our results with calibrated simulations of intermediate-complexity models and 3-D spatially-resolving models. Having said that, we are confident that the discussion you initiated would benefit our paper, and therefore… **Action:** … we will add the above discussion into the text. **Done: p. 15 lines 3-27; p.17 lines 5-38, p.18, lines 22-29**

**Response to Anonymous Referee #3**

Dear Anonymous Referee #3, Thank you very much for your detailed review and helpful suggestions. We appreciate that you consider our findings to be important. The following is our response to your suggestions.

**Suggestion:** This is an interesting contribution to our understanding of the ice ages and the structure of glacial cycles. However, a broader review of ice age dynamics is needed in the introduction and in the wider paper. This will make it more accessible to a wider audience of

Quaternary scientists. For the introduction, some reference to studies of ice age dynamics would be useful. There are obviously lots of papers you can refer to here, such as Imbrie et al. 1993, Paillard 2001 and Lang and Wolff 2011, for example.

**Answer:** We agree that the introduction may be expanded. Currently, it is focused on the unique properties of our model that allows us to use the π-theorem insightfully. At the same time, it does not provide enough background that would allow our readers to better appreciate the importance of the obtained results.

**Action:** We will expand the introduction with a brief review of the state-of-the-art ice age research. **Done: p.9 lines 17-30**

**Suggestion:** In particular, it would be useful if you can explain more clearly and explicitly the wider significance of your findings for understanding the nature of glacial cycles. Your paper is clearly important because it provides a mathematical solution for understanding ice age dynamics whereas other approaches are more qualitative or semi-quantitative (e.g. Hughes and Gibbard, 2018). However, Hughes and Gibbard (2018) found that our understanding of glacial cycles, especially ice dynamics, is not always easily explained by external forcing such as solar radiation, although this does account for 50-60% of glacier change and associated sea level change through glacial cycles. Internal glacier climate dynamics account for the rest of the glacier variations. A complex interplay of various geographical factors was found to be responsible for the asynchronous spatial variation in global glacier dynamics, in both the largest highand mid-latitude ice sheets as well as in smaller mountain ice caps and glaciers at a range of latitudes around the world. Your modelling appears to incorporate ice sheet dynamics only, and the feedbacks associated with this, and does not account for the complexity of the known spatial and temporal glacial patterns. Of course, I don't expect you to solve this in your modelling, but you should make the reader know that you aware of the limitations of your approach.

**Answer:** Your observation, that the ice-sheet dynamics (equations (1) and (2)) is the most comprehensive and most physically substantiated part of the model, is correct. The equation (3) of the "rest-of-the-climate" is ambiguous but its ambiguity allows us to interpret our experiments with increased (or reduced) $\gamma 2$ as experiments with additional feedbacks of different nature (see also our response to Anonymous Referee #2 https://www.earth-syst-dynam-discuss.net/esd-2019-65/esd-2019-65-AC2- supplement.pdf). In other words, we are uncertain about some key mechanisms that we have chosen to describe using the "rest-of-the-climate" linear equation. Among others, non-linear effects related to the carbon cycle, non-linear effects related to sea-level destabilization of ice sheets and related synchronization, non-linear effects related to atmospheric circulation, or non-linear effects related to biogenic calcifiers and their action on alkalinity, etc. A challenger might thus claim that these effects are so important that they should be taken off $\gamma 2$ and be considered more explicitly. However, we have the hope that even after accounting for these processes, we might end up with a model that still has grossly the same mathematical structure as the Verbitsky et al (2018) model, even though the meaning of some of the variables will have changed.

**Action:** We will add the above discussion into the text. **Done: p. 15 lines 3-27**

**Suggestion:** You conclude that only two factors define most of the ice age dynamics: a) a balance between intensities of climate positive and ice sheet negative dynamics and b) the period T and the amplitude of the external forcing. I can see how for b) this can be constrained from orbital parameters but the variables for a) are potentially very complex and only partially accounted for in your modelling. From this, if we can be confident about b) it would be useful to

see a statement on the comparable effects of a) versus b). You may already do this, but I would like to see a much clearer statement on this matter. For example, be much clearer about the implications of what you mean by "the amplitude and duration of glacial cycles is governed by a property of scale-invariance that does not depend on the underlying positive and negative feedbacks incorporated by the system". Unless you make the wider significance your findings more explicit, then it will have a limited audience. I think the findings are potentially very important, and you need to communicate these more effectively with those researching ice age dynamics, who will then be able to refer to your work, thereby increasing the academic impact of this paper.

**Answer:** Unlike many models of the ice-age climate that postulate internal 100-kyr oscillator, in our model, 100-kyr cycle is produced as a non-linear system response to the astronomical forcing. Nothing happens without astronomical forcing and nothing happens without system internal dynamics. Therefore, it is not possible to quantify precisely the impact of the astronomical forcing versus internal climate dynamics. Furthermore, a similar system response may be observed with different forcing and different internal climate dynamics. For illustration, let us consider equation (7) $P=T\Psi(V,\varepsilon/a)$ and equation (9) $S0=\varepsilon\hat{\ }2\, P\hat{\ }2\, \Phi(V,\varepsilon/a)$.

Here the V-number is defined by the climate dynamics, $\varepsilon/a$ is the relative intensity of the astronomical forcing, T is the forcing period, P is the period of system response, and S' is the amplitude of the system response. Function $\Psi(V,\varepsilon/a)$ defines a forcing-period doubling domain and it may be the same (let say, $\Psi=2$) for different combinations of V and $\varepsilon/a$ (see also Fig. 1). We can only say that the obliquity-period doubling requires both well-developed positive feedbacks in the system ($0.6 < V < 0.8$) and relatively high climate sensitivity to the astronomical forcing ($\varepsilon/a \approx 1$). Moreover, different sets of parameters may lead to the same V-number. Function $\Phi(V,\varepsilon/a)$ and function $\Phi'(V,\varepsilon/a)$ corresponding to the same value of the V-number but formed by the different parameters may not be the same. Similarly, function $\Psi(V,\varepsilon/a)$ and function $\Psi'(V,\varepsilon/a)$ may differ. Most remarkably though, the power degree "2" of the response-period in the equation (9) is defined by the fundamental dimensionality requirements and does not depend on the underlying physics. This is what gives us the property of scale invariance but at the same time makes our efforts to disambiguate historical records even more challenging. As a result, we can't 
[revised manuscript text omitted]

---

## Author Response (AR2)

Dear Dr. Messori,

Thank you for your decision. We have adopted two minor revisions that you and two anonymous reviewers suggested.

**Report #2 (Anonymous Reviewer #1):**

I appreciate the authors' effort in addressing my comments. I am happy with the responses and edits, except regarding my first comment.

At this point, the text that is added to lines 15-18 of Page 6 seems contradictory to what is discussed in lines 13 and 14. If changing one parameter (\epsilon) leads to the expression shown at the end of Line 18, then why would not you except that this expression to hold in general? What matters in dimensional analysis is the change in the dimensionless groups, not a specific parameter. It would be helpful if you further clarify the text that follows Eq. 13.

**Answer/Action: Done: p.6, lines 14-18**

**Report #3 (Anonymous Reviewer #2):**

In my first review I wrote:

"Nonetheless, the robustness of the V number, at least when subject to a simple forcing, is impressive. Therefore, in order to strengthen the paper, and add more new material, my suggestion is that the authors will take this model and add several potential feedback mechanisms to obtain different variation of dynamical systems. Then they will have different V numbers for the different models. If for the same value of the different V numbers, for the different models, the dynamic response of the different models will be similar this will be highly cool and much more robust. This will mean that what truly matters is the ratio between positive to negative feedback mechanisms, not only within the same model but also with similar models of the same family."

I think that the authors' response for this suggestion is only partial and not truly satisfying to evaluate the robustness of the V number in different models.

**Non-public comments to the Author:**

I would specifically encourage you to find a reasonable balance between addressing the point raised by reviewer #3 and not having to carry out major additional analysis.

**Answer/Action:** You are correct in your assessment that a major additional research is needed to provide a definitive answer about the robustness of the *V*- number in different models of ice-ages.

In a mean time, we observe that Figure 1 is consistent with a similar map produced by a conceptual model built on a completely different principle, i.e., a simple oscillator type model of Daruka and Ditlevsen (2016). In both cases, the obliquity period doubling requires well-articulated external forcing in combination with the relatively high *V*-number (or reduced damping in the case of Daruka and Ditlevsen, 2016). This similarity implies that the importance of the *V*-number for a climate system dynamics may extend well beyond the Verbitsky et al (2018) model.

Though in his report the Anonymous Reviewer #2 recommends accepting our paper "as is", we think that an additional paragraph describing the results of Daruka and Ditlevsen (2016) may be helpful.

**Done**: p. 4, lines 10-15

Mikhail Verbitsky and Michel Crucifix

February 3, 2020

[revised manuscript text omitted]